# The Potential Importance of CXCL1 in the Physiological State and in Noncancer Diseases of the Cardiovascular System, Respiratory System and Skin

**DOI:** 10.3390/ijms24010205

**Published:** 2022-12-22

**Authors:** Jan Korbecki, Agnieszka Maruszewska, Mateusz Bosiacki, Dariusz Chlubek, Irena Baranowska-Bosiacka

**Affiliations:** 1Department of Biochemistry and Medical Chemistry, Pomeranian Medical University in Szczecin, Powstańców Wlkp. 72, 70-111 Szczecin, Poland; 2Department of Physiology and Biochemistry, Institute of Biology, University of Szczecin, Felczaka 3c, 71-412 Szczecin, Poland; 3Department of Functional Diagnostics and Physical Medicine, Pomeranian Medical University in Szczecin, Żołnierska 54, 71-210 Szczecin, Poland

**Keywords:** CXCL1, CXCR2, heart, liver, chemokine, cytokine, neutrophil, CINC-1, KC, Gro-α

## Abstract

In this paper, we present a literature review of the role of CXC motif chemokine ligand 1 (CXCL1) in physiology, and in selected major non-cancer diseases of the cardiovascular system, respiratory system and skin. CXCL1, a cytokine belonging to the CXC sub-family of chemokines with CXC motif chemokine receptor 2 (CXCR2) as its main receptor, causes the migration and infiltration of neutrophils to the sites of high expression. This implicates CXCL1 in many adverse conditions associated with inflammation and the accumulation of neutrophils. The aim of this study was to describe the significance of CXCL1 in selected diseases of the cardiovascular system (atherosclerosis, atrial fibrillation, chronic ischemic heart disease, hypertension, sepsis including sepsis-associated encephalopathy and sepsis-associated acute kidney injury), the respiratory system (asthma, chronic obstructive pulmonary disease (COPD), chronic rhinosinusitis, coronavirus disease 2019 (COVID-19), influenza, lung transplantation and ischemic-reperfusion injury and tuberculosis) and the skin (wound healing, psoriasis, sunburn and *xeroderma pigmentosum*). Additionally, the significance of CXCL1 is described in vascular physiology, such as the effects of CXCL1 on angiogenesis and arteriogenesis.

## 1. Introduction

Intercellular communication is an essential function in a multicellular organism and involves the cellular secretion of a variety of factors into the extracellular space. This communication is the basis for the functioning of the immune system, in which a significant role is played by cytokines [1]. This group of regulatory proteins includes chemokines, i.e., chemotactic cytokines that serve as chemoattractants for the cells of the immune system. Their main function is to direct the migration of immune cells to target sites. Almost 50 representatives of chemokines have been described so far [2], divided into four sub-families depending on the conserved cysteine motif at the N-terminus. One of them is the group of CXC chemokines.

One of the CXCL1. One of its first described properties was its ability to stimulate melanoma proliferation, hence, leading to one of its first names, i.e., melanoma growth stimulating activity (MGSA) [3]. CXCL1 was also initially classified as an ELR^+^ CXC chemokine, i.e., one of the seven CXC chemokines that are ligands of CXCR2, the main receptor for CXCL1.

As the expression of CXCR2 occurs on vascular endothelial cells [4,5], its activation causes endothelial cell migration and consequent angiogenesis [4,5]. CXCR2 expression also occurs in neutrophils [6,7,8] and therefore the main and most important property of CXCL1 is attracting neutrophils. Increasing CXCL1 expression at sites involved in either inflammatory or disease reactions results in infiltration by neutrophils. Factors that increase CXCL1 expression are pro-inflammatory cytokines such as interleukin-1β (IL-1β) and tumor necrosis factor α (TNF-α) [9,10]. These cytokines activate nuclear factor κB (NF-κB), which increases *CXCL1* gene expression. Another factor is interleukin-17 (IL-17), which increases the stability of CXCL1 mRNA and thus the expression of CXCL1 [11,12,13]. In affected tissues, neutrophils destroy pathogens by phagocytosing them and using reactive oxygen species (ROS), reactive nitrogen species (RNS) and other small-molecule reactive compounds against pathogens [14,15]. Neutrophils also secrete proteases [16] and pro-inflammatory cytokines and thus participate in inflammatory responses [17]. However, uncontrolled neutrophil activity can be dangerous as it leads to tissue damage.

The most significant CXCL1 receptor is CXCR2 [18,19,20]. This receptor is also activated by other CXC chemokines: CXCL2; CXCL3; CXCL5; CXCL6; CXCL7 and CXCL8/interleukin-8 (IL-8) [19,20]. CXCL1 can also activate CXC motif chemokine receptor 1 (CXCR1) but at concentrations approximately 100x greater than for CXCR2 [18]. Therefore, the importance of CXCR1 in the action of CXCL1 is minor. CXCR1 is activated by CXCL8/IL-8 and CXCL6. CXCR2 activation results in signal transduction via multiple pathways [21]. Since CXCR2 is a G protein-coupled receptor, CXCR2 activation causes signal transduction to inhibitory guanine nucleotide regulatory protein (Gα_i_). In particular, Gα_i2_ [22,23] and Gα_i3_ [23] are activated, which results in the inhibition of adenylyl cyclase activity and thus a reduction in cyclic adenosine monophosphate (cAMP) levels in the cytoplasm. G protein also mediates the activation of phosphatidylinositol-4,5-bisphosphate 3-kinase (PI3K) [23,24] and phospholipase C-β (PLC-β) [23,25,26]. PI3K induces protein kinase B (PKB)/Akt activation, while PLC-β is responsible for Ca^2+^ mobilization. On the cytoplasmic side, various proteins important in signal transduction bind to CXCR2, which is collectively known as CXCR2 "chemosynapse" [27,28], responsible for the activation of various signalling pathways, including extracellular signal-regulated kinase (ERK), mitogen-activated protein kinase (MAPK), focal adhesion kinase (FAK), Src, cell division control protein 42 homolog (Cdc42) and actin polymerization [21]. Most of these pathways are significant in cell migration induced by CXCL1.

## 2. Research Methodology

According to PubMed (https://pubmed.ncbi.nlm.nih.gov/ accessed on 1 May 2022), around 5500 articles about CXCL1 have been published. However, there has not been a proper review to summarize the current knowledge of the functions and importance of this chemokine. For this reason, we have embarked on a series of literature reviews on CXCL1. In this paper, we focus on the importance of CXCL1 in non-cancer diseases of the cardiovascular system, respiratory system and skin, based on literature available from the PubMed search engine as of November 25, 2021.

The biggest problem in writing this paper was the lack of an animal model of human CXCL1. Humans have seven CXCR2 ligands, while mice have six CXCR2 ligands [29,30]. All mouse and human CXCR2 ligands activate the CXCR2 receptor. At the same time, the role of a given CXCR2 ligand has been acquired through the evolution of each species. Due to the fact that mice and humans evolved separately, their individual CXCR2 ligands differ. For this reason, it is impossible to equate the role of a given CXCR2 ligand in murine and human disease models. 

In our work, we have been able to partly solve this problem. First, we looked for information on whether there is an upregulation of CXCL1 in the blood or in the affected tissue in a patient with a given disease. If this was the case, we considered that CXCL1 might have some role in the disease. Next, we looked for information on a murine or animal model of the disease. In mice, these models usually show an increase in CXCL1/keratinocyte-derived chemokine (KC) expression. As mouse CXCL1/KC and human CXCL1 activate CXCR2, we equated their role in a given disease model. 

Experimental studies in mice mostly concern CXCR2 inhibitors which inhibit the action of all CXCR2 ligands, thus showing the significance of these ligands in a given disease. Although researchers usually focus on a single CXCR2 ligand (e.g., CXCL1), it cannot be excluded that other CXCR2 ligands are significant in the discussed diseases.

## 3. Cardiovascular System

### 3.1. The Physiology of Blood Vessels

CXCL1 is important in blood vessel function due to its pro-angiogenic properties–promoting the CXCR2-dependent proliferation, migration and tube formation of endothelial cells [31,32]. CXCL1 expression is upregulated by vascular endothelial growth factor (VEGF) [33,34]. Therefore, VEGF and CXCL1 factors participate together in angiogenesis. The described influence of CXCL1 on endothelial cells also implicates CXCL1 in blood vessel regeneration as CXCL1 expression in endothelial cells is increased by tissue factor, the primary initiator of blood coagulation [35]. CXCL1 also causes the recruitment of circulating endothelial progenitor cells [36,37], endothelial cell chemotaxis and proliferation [31,32,38], processes that lead to wound repair.

At the same time, CXCL1 in blood vessels can also cause disease complications. In particular, angioplasty and stenting can lead to neointima formation and to the re-stenosis of blood vessels [39]. This process is dependent on the production of CC motif chemokine ligand 2 (CCL2) and CXCL1 by smooth muscle cells. These chemokines induce the recruitment of Sca-1^+^ vascular stem/progenitor cells to blood vessel sites that have undergone angioplasty and stenting, in processes dependent on CXCR2 and CC motif chemokine receptor 2 (CCR2). The recruited Sca-1^+^ vascular stem/progenitor cells participate in neointima formation [39].

CXCL1 is also important in arteriogenesis, as its expression in vein and aortic endothelial cells is increased during increased blood pressure and thus shear stress [40,41]. This is followed by CXCR2-dependent adhesion of monocytes to endothelial cells [41]. These monocytes differentiate into M2 macrophages [42] which then secrete growth factors that act on smooth muscle cells and endothelial cells, thus enabling collateral remodeling and arteriogenesis [43].

### 3.2. Atherosclerosis

Atherosclerosis is an arterial disease that is characterized by inflammatory reactions in the blood vessel walls, intimal lipoprotein deposition and the formation of atherosclerotic plaques in advanced stages of the disease [44]. Atherosclerosis leads to atherosclerotic cardiovascular disease (CVD) which often ends in death due to myocardial infarction or ischemic stroke (it is estimated that 31% of all deaths worldwide are related to CVD) [45]. In this disease, the development of atherosclerosis involves macrophages. These cells, under the influence of oxidized low-density lipoprotein (oxLDL), are converted to foam cells. 

CXCL1 is another element in the pathophysiology of atherosclerosis. Expression of this chemokine and CXCR2 receptor occurs in humans in the intima of atherosclerotic lesions [46]. oxLDL causes an increase in CXCR2 expression on peripheral blood monocytes [47]. Activated mast cells [48] and macrophages, in particular those interacting with vascular smooth muscle cells [49], produce CXCL1 in intima and perivascular tissue. Another source of CXCL1 in arterial vessel walls is endothelial cells where oxLDL increases the expression of CXCL1 [50]. At the same time, lysophosphatidic acid (LPA) may also be responsible for the production of CXCR2 ligands in arterial vessel walls [51]. LPA induces an increase in the release of endothelial CXCL1, or at least CXCL1/KC, the murine paralog for human CXCL1, as shown by experiments on mice [49]. Pro-inflammatory cytokines such as IL-1β and TNF-α are also important factors that increase CXCL1 expression in the inflamed endothelium and atherosclerotic lesions [52].

CXCL1 causes the recruitment of neutrophils into the atherosclerotic plaque [48]. These cells play an important role in atherosclerotic lesions as the recruited neutrophils secrete pro-inflammatory cytokines that participate in inflammatory responses [53]. Neutrophils also secrete proteases which cause plaque destabilization. CXCL1 also affects monocytes. The CXCL1→CXCR2 axis increases adhesion and arrest of monocytes to the inflamed endothelium, which leads to the recruitment of these cells to the vessel wall [46,50,51,54,55]. The action of CXCL1 in recruiting monocytes to early atherosclerotic lesions seems to be more important than that of CCL2 [54,55]. However, there are also studies contradicting this and showing that the aforementioned axis is only important in advanced atheromatous plaques [56]. CXCL1 also affects monocytes after recruitment. It increases the expression of class A scavenger receptor (SR-A) and cluster of differentiation 36 (CD36) in macrophages [52], both being receptors for oxLDL [57,58]. As CXCL1 increases the accumulation of oxLDL in macrophages, it increases the rate of conversion of these cells to foam cells [52].

CXCL1 also causes the release of matrix metalloproteinases (MMPs) from vascular smooth muscle cells, which leads to plaque destabilization (Figure 1) [52].

Finally, CXCL1 may also participate in atherosclerosis regression [59]. It is responsible for the recruitment of bone-marrow endothelial progenitor cells to atherosclerotic plaque, which leads to plaque resolution.

### 3.3. Atrial Fibrillation

One of the consequences of hypertension is atrial fibrillation [60]. Patients with atrial fibrillation have higher levels of CXCL1 and CXCR2^+^ monocytes in their blood [61]. At the same time, atrial fibrillation patients have lower CXCL1 expression in the atrium heart tissue and higher CXCR2 expression compared to healthy subjects [62]. This suggests that if CXCR2 has a role in the pathogenesis of atrial fibrillation, then it is activated by a ligand other than CXCL1. Animal studies have shown that atrial fibrillation is associated with increased expression of CXCR2 ligands in the atrium and subsequent infiltration by CXCR2^+^ immune cells [61]. In particular, infiltration of monocytes in the atrium tissue leads to an increase in the number of macrophages in the atrium [61]. The level of CXCL1 expression in the atrium in patients with atrial fibrillation is strongly positively correlated with activated mast cells, weakly positively correlated with M2 macrophages and negatively correlated with M1 macrophages [63]. Mast cells may contribute to atrial fibrillation by secreting platelet-derived growth factor A (PDGF-A) [64]. This growth factor increases collagen production by cardiac fibroblasts which leads to atrial fibrosis and atrial fibrillation. At the same time, patients with atrial fibrillation do not experience increased infiltration of atrial tissue by mast cells [65], which indicates that this mechanism may not be significant in atrial fibrillation. In humans, an important role in atrial fibrillation may be played by CXCL7/pro-platelet basic protein (PPBP), a ligand of CXCR2. Expression of this chemokine in the atrium in patients with atrial fibrillation is positively correlated with infiltration of atrial tissue by neutrophils and monocytes [63]. In addition, *CXCL7/PPBP* and *CXCL1* are the hub gene in atrial fibrillation with the highest number of associations [63].

### 3.4. Chronic Oschemic Heart Disease

Patients with chronic ischemic heart disease have elevated levels of CXCL1 expression in the blood [66] and heart [67]. In patients with ischemic heart failure, an increase in cardiac CXCL1 expression may be mediated by T helper type 17 (Th17) cells [67]. CXCL1 acts as a chemoattractant for bone marrow-derived endothelial precursors [68] which participate in vasculogenesis and thus counteract disease progression. Nevertheless, CXCR2-dependent recruitment of immune cells to the heart occurs after infarction [69]. The recruited immune cells cause inflammatory reactions that result in further damage to the heart.

### 3.5. Heart Failure

Elevated blood levels of CXCL1 have been observed in patients with heart failure [70,71]. This increase is correlated with the TT genotype of rs33980500 of TRAF3 Interacting Protein 2 (TRAF3IP2) [70]. In contrast, no increase in CXCR2 expression has been observed in the hearts of patients with heart failure [72]. This shows that elevated levels of CXCL1 in the blood can lead to heart failure as shown by experiments on laboratory animals. In angiotensin II-treated mice, there is an increase in CXCL1/KC expression in the heart [71]. It is a mouse CXCR2 ligand often equated with human CXCL1. The increased expression of CXCR2 ligands in the heart leads to the infiltration of the heart by CXCR2^+^ immune cells, including macrophages, neutrophils, and T cells [61,71]. The same results were obtained by studying spontaneously hypertensive rats [73,74]. In the heart, these immune cells cause cardiac hypertrophy, fibrosis, atrial fibrillation and inflammation [61,69,71]. Mast cellsproduce PDGF-A that causes atrial fibrosis [64]. In addition, macrophages cause atrial fibrosis [75], inflammation in heart tissue and cardiac hypertrophy [76]. These incremental changes in the heart inevitably lead to heart failure [77,78].

### 3.6. Hypertension

CXCL1 can also cause hypertension. People with hypertension have higher levels of CXCL1 in their blood than healthy individuals [79]. The mechanism of its involvement in the pathogenesis of hypertension has been studied in laboratory animals. In mice treated with angiotensin II, the resulting hypertension is dependent on the CXCL1/KC→CXCR2 axis [80]. In this process, increased CXCL1/KC expression in the aorta leads to infiltration of the aorta by cells with a high CXCR2 expression, in particular by macrophages [80,81]. At the same time, the infiltration by monocytes, from which macrophages are formed, may depend on activated B cells [82]. Macrophages have expression of angiotensin type 1 receptor (AT1R) and angiotensin type 2 receptor (AT2R) and for this reason can respond to angiotensin [83,84]. Among other things, angiotensin II causes an increase in toll-like receptor 4 (TLR4) expression which leads to an increase in pro-inflammatory cytokine production in macrophages, which results in inflammation in blood vessels [84]. Additionally, macrophages in the blood vessel wall under the influence of B cells produce transforming growth factor β (TGF-β), which leads to extracellular matrix remodeling, vascular stiffness and consequent hypertension [84].

Hypertension can cause hypertensive retinopathy. Patients with hypertensive retinopathy have higher levels of CXCL1 in their blood than patients with hypertension but without hypertensive retinopathy and healthy individuals. Studies on mice have shown that angiotensin II-induced hypertension increases CXCL1/KC expression in the retina [79], which increases infiltration of the retina by CXCR2^+^ immune cells, particularly by neutrophils and macrophages. In the retina, these cells produce ROS which leads to oxidative stress. These cells are involved in inflammation, which, together with ROS production, leads to retinopathy [79].

### 3.7. Sepsis

Sepsis is an infection with acute organ dysfunction that often results in death [85]. It is difficult to attribute to a particular organ because sepsis causes symptoms in multiple organs. In the US alone, there are 300 cases of sepsis per 100,000 people per year. We have included sepsis as a cardiovascular disease due to the importance of circulation in the early stages of the sepsis course. Sepsis is closely associated with the inflammatory response which is associated with elevated blood levels of pro-inflammatory cytokines such as TNF-α, IL-1β and interleukin-6 (IL-6) [86]. CXCL1 levels also tend to increase in sepsis patients, with an increase close to statistical significance relative to healthy individuals (*p* = 0.07) [87]. However, in humans with sepsis, the most important CXCR2 ligand is CXCL8/IL-8, whose blood levels are significantly elevated [87]. In sepsis, the CXCR2 ligands have two actions. First, they cause mobilization of neutrophils from the bone marrow and infiltration of various organs by these cells. Then, neutrophils contribute to the eradication of bacterial pathogens [88] but also contribute to tissue damage resulting in organ dysfunction.

One of the symptoms of the described disease state is sepsis-associated encephalopathy—damage to the nervous tissue. In the brains of individuals who died from sepsis, expression of CXCL1 was significant in two out of three reported cases [89], although increased expression of CXCL8/IL-8 was observed in all three cases. Increased production of CXCR2 ligands is also confirmed in animal models [90,91]. CXCL1 production in the brain in sepsis is carried out by astrocytes [90] and microglial cells [92], in a process dependent on pro-inflammatory cytokines. Sepsis-associated encephalopathy is caused by the infiltration of the brain by neutrophils, in a process dependent on the expression of CXCR2 on cerebral endothelial cells [93]. Activation of this receptor results in the increased expression of P-selectin, vascular cell adhesion molecule-1 (VCAM-1) and intracellular adhesion molecule-1 (ICAM-1), which are involved in the rolling and adhesion of leukocytes and the subsequent recruitment of neutrophils to the brain [90]. However, the primary driver of endothelial activation may not be the CXCL1→CXCR2 axis but lipopolysaccharide (LPS) and TNF-α [93].

Sepsis also causes acute kidney injury (AKI). Studies on mice have shown that CXCL1/KC and CXCL2/macrophage inflammatory protein-2 (MIP-2) expression is increased in the kidney during sepsis [94,95,96]. *CXCL1/KC* is a hub gene in septic AKI in mice [96]. The IL-17–dependent increase in murine CXCL1/KC and CXCL2/MIP-2 expression in the kidney leads to the infiltration of the kidney by neutrophils [95] which contribute to kidney damage by secreting proteases, particularly neutrophil elastase [97]. This process occurs in patients with sepsis, as patients with renal dysfunction show an increased neutrophil-to-lymphocyte ratio [98]. Nevertheless, further studies are required to demonstrate which CXCR2 ligands are involved in this process in humans.

Murine sepsis is associated with the purinergic receptor P2X7 (P2X7)-dependent generation of reactive species and lipid peroxidation in the liver, leading to the apoptosis of liver cells [99]. Additionally in the liver, in hepatocytes, there is a P2X7-dependent increase in the expression of CXCL1/KC and CXCL2/MIP-2 as shown by experiments in mice [99,100,101]. This leads to infiltration of the liver by neutrophils which then cause liver injury in a process dependent on apoptotic cells in the liver [101]. CXCL1/KC produced by the liver also causes myeloid-derived suppressor cells (MDSC) mobilization which contributes to the control of systemic inflammation—a mechanism that may mitigate the course of sepsis [102]. Nevertheless, the involvement of CXCL1 in liver dysfunction requires further investigation.

Sepsis often leads to ileus, associated with TLR4 activation on myocytes in the ileum, a process which leads to an increase in CXCL1/KC expression in mice [103]. This chemokine contributes to reduced contraction amplitude in the ileum and consequently to the ileus [104]. Further studies on humans are required to demonstrate which CXCR2 ligand plays an important role in this process.

## 4. Respiratory System

### 4.1. Asthma

Asthma is an inflammatory respiratory disease with various etiologies that affects more than 300 million people worldwide [105]. This disease is also associated with increased levels of CXCL1, for example in the bronchoalveolar lavage fluid [106] or bronchial smooth muscle cells in asthmatic patients [107]. *CXCL1* is a hub gene with the one of the greatest number of associations in asthma [108] which indicates that this chemokine may play an important role in this disease.

CXCL1 expression in asthma depends on IL-17, an interleukin produced in asthmatics by eosinophils and T cells [109]. IL-17 increases CXCL1 expression and enhances the induction of its expression by IL-1β and TNF-α in lung microvascular endothelial cells [110], bronchial epithelial cells [111], eosinophils [112], bronchial fibroblasts [109] and airway smooth muscle cells [113]. CXCL1 expression in bronchial epithelial cells may also depend on the action of mast cells [114].

CXCL1 has multiple functions in the pathogenesis of asthma. Although human eosinophils do not express CXCR1 or CXCR2, meaning CXCL1 has no direct effect on these cells [112], they themselves produce CXCL1 as a result of the action of IL-17 [115], which causes the migration of airway smooth muscle cells via two receptors, CXCR1 and CXCR2 [111,116]. On the other hand, the CXCL1 activation of the Duffy antigen receptor for chemokines (DARC)/atypical chemokine receptor 1 (ACKR1) inhibits the migration of airway smooth muscle cells [116].

CXCL1 is also a pro-angiogenic chemokine which acts on microvascular endothelial cells [107] and is responsible for the recruitment of bone marrow-derived endothelial progenitor cells [117]. This leads to neovascularization and airway remodeling, processes characteristic for asthmatics.

CXCL1 may also alleviate asthma symptoms as it inhibits the chemotaxis of mast cells, an effect dependent on CXCR2 [118]. In healthy individuals, IL-1β, Th1 cytokines, Th2 cytokines and the combination of IL-1β with Th1 or Th2 cytokines, increase the expression of CXCL1 in airway smooth muscle much more strongly than in asthmatics. CXCL1 inhibits mast cell chemotaxis caused by CXCL8 and CXCL10. This process is dependent on CXCR2. Inhibition of mast cell chemotaxis reduces the number of these cells in the lungs. In asthmatics, there is a decrease in CXCL1 production by activated airway smooth muscle [118], which leads to the recruitment of mast cells that play an important role in the pathogenesis of asthma [119].

### 4.2. Chronic Obstructive Pulmonary Disease (COPD)

Frequent exposure of the respiratory system to irritant particles, such as cigarette smoke, smoke from biomass fuels and urban air pollution, causes chronic inflammation in the respiratory system which leads to chronic obstructive pulmonary disease (COPD) [120,121]. It is estimated that over 300 million people have COPD [122] and CXCL1 is one of the components of its pathogenesis.

CXCL1 levels are increased in the sputum and lungs in patients with COPD [123,124,125,126]. This may be related to changes in the methylation of the *CXCL1* gene promoter [127]. Sources of this chemokine include lung fibroblasts [128], bronchial epithelial cells [129], some subsets of alveolar epithelial type II cells [130] and airway smooth muscle cells [131] as the exposure of these cells to irritant particles increases CXCL1 expression. The effect of cigarette smoke on CXCL1 expression in the alveolar epithelial cells depends on the upregulation of plasminogen activator inhibitor-1 (PAI-1) expression by p53 [125]. PAI-1 in alveolar epithelial cells increases the expression of CXCL1 but also CXCL2, CXCR2 and ICAM-1. In addition, the increase in CXCL1 expression in alveolar epithelial cells induced by cigarette smoke may occur due to the activation of NF-κB [132]. At the same time, this effect depends on the cell type. Cigarette smoke does not increase but decreases the production of CXCL1 and other chemokines in macrophages [133,134,135], due to an increase in M2 polarization of alveolar macrophages [134]. On the other hand, cigarette smoke increases CXCL8/IL-8 expression in these cells [133] and increases the production of CCL17 and CCL22 [135]. The described effects of cigarette smoke on macrophages are dependent on the reduced expression of p50 NF-κB and IκBα and increased expression of AP-1 subunits [133].

CXCL1 causes an increase in the number of monocytes and thus macrophages in the respiratory system. In COPD patients, monocytes show increased chemotactic responses to CXCL1 and to other CXCR2 ligands [136], but at the same time show a reduced ability to fight pathogens [121].

Neutrophil counts are also increased in the respiratory system in COPD patients and show a positive correlation with CXCL1 levels. [123]. Neutrophils cause alveolar destruction by secreting serine proteases such as elastase, cathepsin G, proteinase-3, matrix metalloproteinase-8 (MMP-8) and matrix metalloproteinase-9 (MMP-9) [121].

### 4.3. Chronic Rhinosinusitis

Chronic rhinosinusitis, involving chronic inflammation of the paranasal sinuses, is a disease that affects between 1% and 5% of the population [137]. Neutrophils play an important role in the course of this disease—they participate in inflammatory reactions by secreting pro-inflammatory cytokines [138]. For example, they cause tissue fibrosis by secreting transforming growth factor β2 (TGF-β2). Neutrophils in chronic rhinosinusitis are recruited to the nasal mucosa by CXCL1. At the same time, neutrophils themselves are an important source of CXCL1 in this disease and thus recruit more neutrophils [139].

CXCL1 expression also occurs in nasal mucosa-derived fibroblasts, in a process dependent on IL-17A [138], thromboxane A_2_ (TxA_2_) and thromboxane receptor (TP) activation on these cells [140]. Another important source of CXCL1 are nasal epithelial cells, which secrete this chemokine as a result of protease-activated receptor-2 (PAR-2) activation in these cells [141]. This has important implications in the bacterial etiology of chronic rhinosinusitis. In this disease, *Staphylococcus epidermidis* secretes serine proteases that activate PAR-2 [142], which leads to an increase in CXCL1 expression [141,142]. Another factor that increases CXCL1 expression in dispersed nasal polyp cells is truncated interleukin-36γ (IL-36γ) [139]. IL-36γ is activated by proteolytic truncation involving neutrophil-derived elastase—this means that neutrophils increase their own recruitment via elastase and IL-36γ.

### 4.4. Coronavirus Disease 2019 (COVID-19)

Severe acute respiratory syndrome coronavirus-2 (SARS-CoV-2) is a virus that caused a global pandemic that was still ongoing at the time of writing this article (late 2021) [143]. CXCL1 is an important factor in the course of this disease. SARS-CoV-2 activates the IL-17 signaling pathway through the interaction of the viral open reading frame 8 (ORF8) protein with interleukin-17 receptor (IL-17R) [144]. Additionally, the SARS-CoV-2 spike protein, depending on TLR2 and NF-κB, increases the expression of CXCL1 as shown by an experiment on macrophages (Figure 2) [145]. As a result of the SARS-CoV-2 action, there was an increase in the expression and secretion of CXCL1 by cells in contact with the virus [146]—one of the components of the cytokine storm caused by SARS-CoV-2. For this reason, researchers have observed increased CXCL1 levels in the bronchoalveolar lavage (BAL) fluids of patients with this disease [147,148]. In the blood of patients with moderate and severe coronavirus disease 2019 (COVID-19) there are elevated levels of CXCL1 and CXCL8/IL-8 [149]. At the same time, the level of these chemokines is dependent on the SARS-CoV-2 variant. Moderate and severe infection with Wuhan and Alpha variants of SARS-CoV-2 cause an increase in CXCL1 levels in the blood, while SARS-CoV-2 Delta and Omicron variants do not cause an increase in CXCL1 levels. Moderate and severe COVID-19 caused by all variants cause an increase in CXCL8/IL-8 levels in the blood [149]. Importantly, CXCL1 and CXCL8/IL-8 are not the only chemokines significant in the course of COVID-19. The disease is also associated with the elevated blood levels of other chemokines such as CCL2/MCP-1, CCL3/MIP-1α, CCL4/MIP-1β, CXCL9/MIG, CXCL10/IP-10 and CX3CL1/Fractalkine, which shows that CXCL1 is only one of many COVID-19 factors. As CXCL1 is a chemoattractant for neutrophils, an increase in CXCL1 levels in the blood causes the mobilization of neutrophils from the bone marrow [150], which causes an increase in the neutrophil–lymphocyte ratio (NLR) values in the blood [151]. CXCL1 also causes the recruitment of neutrophils to the lungs and so an increase in respiratory CXCL1 levels leads to an increase in neutrophils in the BAL fluid of patients with COVID-19 [148]. Subsequently, SARS-CoV-2 causes neutrophil extracellular traps (NETs) to be released by neutrophils in the respiratory system and in the blood [152,153]. As NETs cause damage to organs including lung epithelial cells [152], high neutrophil mobilization, and more specifically a very high NLR value, may be an early predictive factor to critical illness for patients with COVID-19 [154]. In addition, SARS-CoV-2 induces an elevated expression of CXCL1 in cardiomyocytes [155], which leads to the infiltration of the heart by immune cells [61,69,71] and inflammation that can damage the heart. In this process, CXCL1 and CCL2 may act together [155].

### 4.5. Influenza

Influenza is a viral disease of the respiratory tract [156] caused by influenza A virus (IAV), influenza B virus (IBV), influenza C virus (ICV) or influenza D virus (IDV) [156,157,158]. In humans, IAV is the most significant influenza virus as it can cause severe symptoms and pandemics every few years. IDV has been identified relatively recently in cattle but so far there is no data that this virus infects humans [158]. Influenza viruses belong to the family *Orthomyxoviridae* [156,157]. Their genetic material is segmented single-stranded RNA with a negative orientation. In IAV and IBV, the genome is divided into eight segments, and the ICV genome is divided into seven segments [159]. Influenza viruses infect humans through the droplet route and through virus-infected surfaces such as doorknobs [156,157]. The incubation period is short, up to 3 days, and symptoms last up to 7 days. It is estimated that during flu seasons, 10% of the population becomes infected with influenza viruses annually with half a million deaths each year [156]. CXCL1 may play a significant role in the course of influenza. However, all experimental studies of the importance of CXCL1 in influenza are based on mouse studies. Therefore, confirmation of the cited findings in a human model is required.

Studies on mice have shown that IAV causes an increase in the expression and secretion of chemokines such as CCL2, CC motif chemokine ligand 5 (CCL5) and CXC motif chemokine ligand 10 (CXCL10) in infected mouse-lung epithelial cells [160,161]. Additionally, IAV infection results in increased expression of chemokines for neutrophils: CXCL1/KC and CXCL2/MIP-2 in mouse lung epithelial cells (Figure 3) [160,161]. Other sources of CXCL1/KC in IAV infection include dendritic cells, neutrophils and macrophages [162]. At least in mouse CXCL1/KC, expression of this chemokine in dendritic cells, neutrophils and macrophages depends on an increase in transcript stability for CXCL1/KC by NLR family pyrin domain containing 12 (NLRP12) [162]. CXCL1/KC and CXCL2/MIP-2 result in the recruitment of neutrophils to the lungs and subsequent viral clearance and control of the infection [163]. However, influenza infection may result in insufficient or excessive body responses. Too low expression of CXCL1/KC and CXCL2/MIP-2 prevents the immune system from fighting IAV which results in death [164]. On the other hand, an overly high immune response can lead to lung damage which is also fatal [160,162].

IAV, or more precisely the NS1 of IAV, decreases the immune response in the lung to infection with this virus and thus decreases the expression of CXCL1/KC and CXCL2/MIP-2 there [165]. This reduces influenza symptoms but also increases IAV viral load in the lung. In contrast, estrogens increase the expression of CXCL1/KC and CXCL2/MIP-2 in the murine lung [164] and so increase neutrophil recruitment to the lung in females and thus increase viral clearance. Nevertheless, an excessive immune system response leads to lung damage and death, which occurs in older adults [160]. This is associated with the senescence of alveolar epithelial cells which secrete elevated levels of CXCL2/MIP-2, and under IAV infection show higher expression of CXCL1/KC and CXCL2/MIP-2 than young cells. This may explain the higher influenza mortality of people over 65 years old [156].

The aforementioned CXCR2 ligands may play an important role in post-influenza complications. Simultaneous exposure to IAV and *Aspergillus fumigatus* spores reduces CXCL1/KC and CXCL2/MIP-2 expression in mice [166]. As neutrophils are also involved in combating this fungus, lung aspergillosis can occur after the influenza is cured. In addition, neutrophils recruited during influenza in the lungs may be part of an excessive immune response during post-influenza complications. An example of this is *Streptococcus pneumoniae* infection during influenza [163], where neutrophils contribute to excessive lung damage which can result in death. For this reason, inhibitors of CXCR2, the receptor for CXCL1/KC and CXCL2/MIP-2, are suggested to be used in treatment to reduce the action of the described ligands for this receptor.

### 4.6. Lung Transplantation and Ischemic-Reperfusion Injury

During transplantation, the organ transferred to the donor is disconnected from blood circulation for a period of time, which results in an ischemic state. The circulation is then restored during transplantation, in a process known as reperfusion. However, reoxygenation leads to tissue damage in the ischemic-reperfusion injury process. Transplanted lungs are exposed to increased secretion of IL-17 from invariant natural killer T (iNKT) cells and TNF-α from macrophages [167]. These cytokines have been shown to increase CXCL1/KC expression in alveolar type II epithelial cells in mice. CXCL1/KC then causes the recruitment of neutrophils to the lungs and thus results in damage to the transplanted organ [167]. Although CXCL1/KC is a paralog of human CXCL1, their role is not the same. Lung transplant patients have increased expression of CXCL3, CXCL7 and CXCL8/IL-8 in the lung but not CXCL1 [168]. CXCL3, CXCL7 and CXCL8/IL-8 are chemokines that, similar to CXCL1, activate CXCR2 and cause neutrophil recruitment. In humans, these chemokines, and not CXCL1, play important roles in lung ischemic-reperfusion injury [168].

Ex vivo lung perfusion is used to counteract the ischemic-reperfusion injury of transplanted lungs and prolong the time in which lung transplantation is possible. At the same time, after several hours of lung perfusion, there is a decrease in the number of neutrophils in the lungs. There is also an increase in the production of pro-inflammatory cytokines such as IL-1β and TNF-α as well as chemokines such as CXCL1, CXCL2 and CXCL8/IL-8 in the lungs [169]. This indicates that the increased expression of cytokines in the lung may be a significant influence on complications following lung transplantation.

### 4.7. Tuberculosis

Tuberculosis is caused by *Mycobacterium tuberculosis*. The most common manifestation of the disease is pulmonary tuberculosis although tuberculosis in extrapulmonary sites occurs in as much as 15% of infections [170]. The main route of infection is the droplet route. An estimated 1.3 billion people worldwide are infected with *M. tuberculosis* but only some will develop symptomatic tuberculosis. An estimated 9 million people a year develop tuberculosis and 1.5 million a year die from the disease [170]. This is due to risk factors that are associated with immunosuppression, especially human immunodeficiency virus (HIV) infection, malnutrition, immunosuppression associated with organ transplantation and cancer. The highest incidence of tuberculosis is observed in low-income and middle-income countries.

One of the components of tuberculosis pathogenesis is CXCL1. Blood CXCL1 levels are elevated in patients with pulmonary tuberculosis and these levels correlate with bacterial burdens [171,172,173]. CXCL1 can be considered a marker of pulmonary tuberculosis, distinguishing it from latent infection [172]. High blood CXCL1 levels indicate that the patient will show a good response to treatment with anti-tuberculosis drugs [174]. Patients with tuberculous uveitis show higher levels of CXCL1 in aqueous humor compared to healthy individuals [175].

Studies on mice have shown that the interferon α and β receptor subunit 1 (IFNAR1) in *M. tuberculosis* infection increases the expression of pro-inflammatory cytokines as well as CXCR2 ligands [176]. The increased expression of CXCR2 ligands in the lung is associated with increased rates of cell death [176] and IL-17A production by Th17 cells [177]. Increased expression of CXCR2 ligands leads to the recruitment of neutrophils to the lung. As these cells are infected by *M. tuberculosis,* they are an important element of susceptibility to *M. tuberculosis* [176,178]. By secreting CXCL9, they participate in the formation of granuloma [177,179], structures composed of immune cells [180], which constitute the proliferation site for *M. tuberculosis* and facilitate dissemination of this pathogen.

*M. tuberculosis* can infect human dendritic cells [181]. Studies on *Mycobacterium bovis* Bacillus Calmette–Guérin (BCG) and mouse dendritic cells have shown neutrophils play a role in helping the pathogen survive [182]. Infected dendritic cells produce CXCL1/KC and CXCL2/MIP-2 which leads to the recruitment of neutrophils in the vicinity of the infected dendritic cells. Subsequently, through direct contact, especially depending on CD11b integrin, dendritic cells increase the expression and secretion of IL-10 by neutrophils. This cytokine decreases IL-17A production by Th17 CD4^+^ cells which inhibits immune mechanisms against *M. tuberculosis*.

*M. tuberculosis* also infects macrophages [183]. The TLR2-dependent activation of mammalian sterile 20-like 1 and 2 kinases (MST1/2) in infected macrophages causes the activation of interferon regulatory factor 3 (IRF3), which leads to an elevated expression and secretion of CXCL1 and CXCL2. These chemokines increase the expression of pro-inflammatory cytokines, NADPH oxidase, inducible nitric oxide synthase (iNOS) and β-defensins in macrophages, which leads to an immune response against *M. tuberculosis* [183,184].

## 5. Skin

### 5.1. Wound Healing

The skin is the organ that separates the body from the external environment. It protects the body from drying out, protects against minor mechanical damage and provides a barrier to pathogens. For this reason, any damage to the skin should be repaired immediately. During wound healing, an important role is played by chemokines, including CXCL1 whose expression increases significantly in these instances [185,186] but also other chemokines, including CC motif chemokine ligand 14 (CCL14), CC motif chemokine ligand 27 (CCL27) and CXCL10 [186]. The increase in CXCL1 expression appears to be much smaller than that of other chemokines. However, this effect may depend on the type of skin undergoing repair. Oral keratinocytes, relative to dermal keratinocytes, produce much more CXCL1 in response to pro-inflammatory cytokines [187]. In addition, CXCL1 expression is regulated by tissue hormones in the skin. During wound healing, CXCL1 expression is decreased by dermokine-β, which is expressed in marginal keratinocytes [188].

CXCL1 is an important component during skin repair. CXCL1 causes the recruitment of neutrophils in the inflammatory stage of wound healing [188], which leads to the removal of pathogens in damaged skin. CXCL1 also causes proliferation [185,186] and migration [186] of keratinocytes during the proliferative phase of wound healing. Other studies have shown no effect of CXCL1 on the migration of these cells [185]. The described processes lead to re-epithelialization and wound closure [186]; however, other chemokines are also responsible for this process [186].

### 5.2. Psoriasis

Psoriasis is a chronic inflammatory skin disease associated with skin lesions, most often plaques with reduced pigmentation, resulting in a pink or ashy color [189]. Approximately 60 million people worldwide are affected by this disease with the prevalence depending on the region [189]. The highest prevalence is found in Australasia, at 1.99% [190], followed by Western European and North American countries, at 1.92% and 1.5%, respectively. In contrast, in East Asian countries, only 0.14% of the population has psoriasis [190].

CXCL1 is one of the factors involved in the inflammatory responses that are responsible for skin lesions in psoriasis; other factors include the IL-23→Th17 cells axis, TNF-α and interferon-γ (IFN-γ) [189]. The expression of CXCL1 is elevated in psoriatic lesions [191,192]; however, it occurs normally in keratinocytes in the suprapapillary layer and in vessels in the papillary dermis [193,194]. Serum levels of CXCL1 are elevated in patients with psoriasis [195,196]. In addition, in generalized pustular psoriasis, serum CXCL1 levels are higher than in psoriasis vulgaris and are proportional to severity scores [196,197]. Lastly, the *CXCL1* gene has been identified as a hub gene for mild psoriasis [198]. All the aforementioned information makes the case for a strong association between CXCL1 and psoriasis.

The elevated expression of CXCL1 in keratinocytes and melanocytes is due to the action of both IL-17 and TNF-α, cytokines whose expression is also elevated in psoriatic lesions [199,200,201,202,203]. The synergistic effect of these two cytokines may account for the increased expression of CXCL1. Another factor that increases the effect of IL-17 on CXCL1 expression in keratinocytes may be the activation of protease-activated receptor 2 (PAR2) [201], although that effect may be offset by the active form of vitamin D [201]. The expression of CXCL1 may also be triggered by IL-36γ [204,205], a cytokine that is synthesized in an inactive form and then processed by cathepsin G, a protease secreted by neutrophils. This means that neutrophils also increase their own recruitment by truncated IL-36γ, which increases CXCL1 expression. CXCL1 expression in keratinocytes may also be due to titers in miRNA expression. The expression of miRNA-31 is increased in keratinocytes in psoriatic lesions [206]. As miRNA-31 can reduce the expression of serine/threonine kinase 40 (STK40) which in turn decreases the activity of NF-κB, an increase in miRNA-31 expression increases the activity of NF-κB, a transcription factor responsible for the expression of pro-inflammatory genes and CXCL1.

Cytokines IL-17 and TNF-α cause an increase in β-defensin 3 expression in psoriatic lesions, particularly in keratinocytes (Figure 4) [202]. As β-defensin 3 is an antagonist for melanocortin-receptor 1, this mechanism inhibits the production of the skin pigment melanin by melanocytes. At the same time, CXCL1 is a mitogen for melanocytes [202,207,208] and therefore psoriatic lesions have an elevated number of melanocytes that do not produce pigment. This is connected with the inhibition of melanogenesis by CXCL1 [209]. Due to the lack of melanin, psoriasis lesions have a characteristic discoloration. The use of treatments in patients with psoriasis reduces the level of TNF-α in psoriatic lesions which allows melanocytes to produce pigment. Because there are many more of these cells in psoriatic lesions than in the healthy skin, treatment results in the hyper-pigmentation of these areas [202].

The areas in the skin with high CXCL1 expression experience neutrophil infiltration [193,210]. As CXCL1 expression occurs in colocalization with CXCL8/IL-8 [193,194], both these chemokines may be involved in neutrophil infiltration. Neutrophils take part in the inflammatory response in psoriatic lesions [203,210].

Neutrophils produce and secrete ROS and other reactive low molecular weight compounds that damage macromolecules in the skin [211,212]. In patients with psoriasis, neutrophils have a greater capacity for respiratory burst [211]. Together with mast cells, neutrophils are the main source of IL-17 in psoriasis lesions [203,213,214,215], a cytokine that increases the expression of chemoattractants for neutrophils, including CXCL1 in keratinocytes. In this way, neutrophils enhance their own recruitment [199,200,201,203]. IL-17 is also crucial in inflammatory responses in psoriasis lesions [189] which has resulted in the development of drugs targeting this cytokine.

Upon entering the skin, neutrophils form NETs [214], structures that increase the activation of Th17 cells [216]. NETs also increase inflammatory responses in the skin by enabling TLR4 and IL-36R crosstalk [217] and form a complex with LL-37 [218], a skin antimicrobial peptide [219] produced by keratinocytes in psoriasis lesions [220]. The complex of LL-37 and NET activates TLR8/TLR13 which results in increased production and secretion of pro-inflammatory cytokines by neutrophils as well as the formation of NETs by other neutrophils, which drives inflammatory responses in psoriasis lesions [218].

Neutrophils also secrete proteases that are involved in the development of psoriasis lesions [210]. These enzymes activate ligands for EGFR, which leads to excessive proliferation of keratinocytes [210,221]. Lastly, neutrophil proteases, especially cathepsin G, activate IL-36γ [205], a cytokine important in inflammatory responses in psoriatic lesions [222,223].

Studies in mice have shown that CXCL1 may also be important in the development of psoriatic arthritis. In psoriatic arthritis, γδ T cells play a significant role in the response to the disease [224]. These immune cells increase the expression of CXCR2 ligands in joints, which is followed by infiltration by neutrophils, which have a destructive effect and lead to the development of psoriatic arthritis [224].

### 5.3. Sunburn

Sunburn is the result of excessive skin exposure to ultraviolet (UV) light. It mainly occurs under the influence of too much sunbathing and staying in the sun. This is followed by an increase in the expression of chemokines responsible for recruiting neutrophil, monocyte and dendritic cells to the skin. In particular, there is an increase in the expression of CCL2, CCL20, CXCL1 and CXCL8/IL-8 [225]. Studies on mice have shown that keratinocytes and fibroblasts are responsible for the expression of CXCL1/KC, a paralog for human CXCL1 [226]. At the same time, CXCL1/KC expression was not TNF-α dependent. It is likely that in human skin the expression of CXCL1 under UV light occurs in keratinocytes and fibroblasts, but this needs to be verified experimentally. CXCL1 causes the recruitment of neutrophils to skin previously exposed to UV light [225]; these cells are involved in inflammatory reactions in the skin that can promote skin cancer formation, known as photo-carcinogenesis [226].

### 5.4. Xeroderma Pigmentosum

*Xeroderma pigmentosum* is an autosomal recessive disease [227]. The incidence of this disease is population–dependent. The prevalence of *xeroderma pigmentosum* is estimated to range from 1 per 100,000 in the United States to 35 per 100,000 in Japan. It is characterized by defects in DNA repair enzymes, specifically the nucleotide excision repair (NER) pathway [227]. Because of this, UV radiation, a component of sunlight, causes DNA damage in the skin that is not efficiently repaired in individuals with *xeroderma pigmentosum*. This leads to skin lesions in which a certain role is played by CXCL1. During skin exposure to UV light, xeroderma pigmentosum group A (XPA) protein-deficient mice show an increase in CXCL1/KC expression [228], a paralog for human CXCL1. CXCL1/KC causes infiltration of the skin by neutrophils which participate in disease reactions in the skin, including DNA damage by producing ROS [228]. However, the presence of increased CXCL1 expression in the skin requires confirmation in humans.

### 5.5. Itchy Skin

CXCR2 ligands can also cause itchy skin, as confirmed by studies in rats. However, further research is required on the relationship between itchy skin and CXCL1 in humans.

An increase in CXCR2 ligands in the skin leads to the activation of transient receptor potential vanilloid type 1 (TRPV1) located on dorsal root ganglia neurons [229,230]. The mechanism of TRPV1 activation by CXCR2 ligands is dependent on CXCR2 activation on dorsal root ganglia neurons [230]. This is followed by signal transduction on actin filaments and TRPV1 activation, which manifests as an itchy sensation. 

## 6. Perspectives for Further Research

The importance of CXCL1 in physiology and in diseases has been very well understood. Nevertheless, the greatest difficulty in understanding the action of CXCL1 is the lack of an animal model to study human CXCL1, as the CXCR2 ligand system differs between humans and rodents. This is related to the nature of evolution [29,30]. The ancestor of today’s mammals had far fewer CXCR2 ligand genes; over millions of years, a single CXCR2 ligand gene underwent multiple duplications. Subsequently, individual representatives of CXCR2 ligands took on the closely matched but no identical functions that they currently perform in the human, mouse or rat body. For this reason, mouse CXCL1/KC and rat CXCL1/CINC-1 are not strictly equivalent to human CXCL1. Human CXCL1 does often have the same function as mouse CXCL1/KC or rat CXCL1/CINC-1 but not always. There are reports where an in vivo results obtained from rodents contradicted the results obtained from human material [168,231,232]. For this reason, results obtained from laboratory animals should be taken with great caution.

Very often, either in physiology or in diseases, a pair of CXCR2 ligands complement each other. However, most experimental work examines only one CXCR2 ligand without analyzing other chemokines. For this reason, future work should focus on a more extensive examination of either physiological or disease mechanisms, and in particular should examine the significance of a given CXCR2 ligand in the context of other chemokines.

Another direction of research into the significance of CXCL1 in diseases should be the development of therapeutic approaches that target this chemokine. An example of such a therapy could be the use of anti-CXCL1 antibodies [233]. As CXCL1 may act together with other CXCR2 ligands, in particular CXCL8/IL-8 [234], a better therapeutic approach may be to use inhibitors of this receptor [235,236] or the inhibitor of CXCR1/CXCR2 [237,238], which inhibits not only the receptor for CXCL1 and CXCL8/IL-8 but also the other receptor for CXCL8/IL-8. CXCL1 plays a significant role in the early stages of some diseases [239], causing neutrophil infiltration, which damages the tissue and aggravates symptoms. Although reducing the expression of CXCL1 can only inhibit disease progression, combining this therapeutic approach with currently used drugs seems to be a promising approach.

## 7. Conclusions

The main conclusions of this paper are as follows: knowledge on the role of CXCL1 in the non-cancerous diseases discussed in this paper is incomplete due to the lack of an animal model of the human CXCR2 ligand system; the currently applied therapies do not target CXCL1 or its receptor, CXCR2.

## Figures and Tables

**Figure 1 ijms-24-00205-f001:**
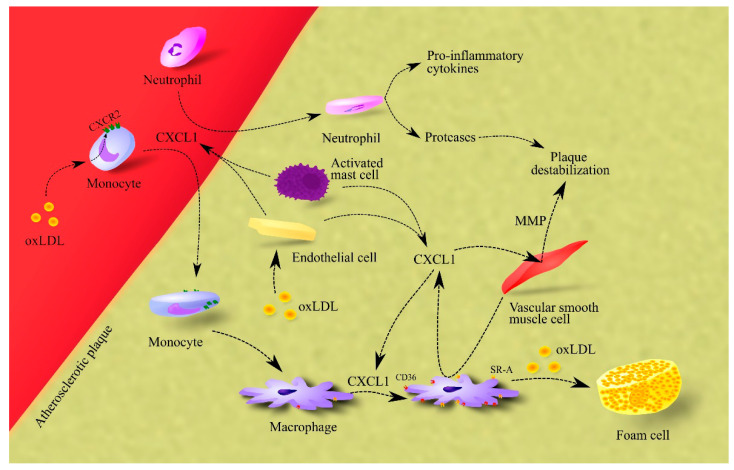
Importance of CXCL1 in atherosclerosis. In atherosclerotic plaque, endothelial cells, macrophages and activated mast cells are responsible for CXCL1 production. This chemokine causes the recruitment of neutrophils and monocytes to atherosclerotic plaque. The effect on monocytes is enhanced by oxLDL, which increases CXCR2 expression on these cells. After recruitment, neutrophils secrete pro-inflammatory cytokines that increase inflammation in the atherosclerotic plaque. These cells also secrete proteases that cause plaque destabilization. In turn, monocytes differentiate into macrophages which, under CXCL1, increase the expression of CD36 and SR-A, receptors for oxLDL. Therefore, CXCL1 increases the uptake of oxLDL by macrophages. This elevates the rate of foam cell formation in the atherosclerotic plaque. Additionally, CXCL1 causes an increase in the production and secretion of MMPs in vascular smooth muscle cells; these are proteases that cause plaque destabilization. Abbreviations: CXCL1–CXC motif chemokine ligand 1; CXCR2–CXC motif chemokine receptor 2; MMP–matrix metalloproteinases; oxLDL–oxidized low-density lipoprotein; SR-A–class A scavenger receptor.

**Figure 2 ijms-24-00205-f002:**
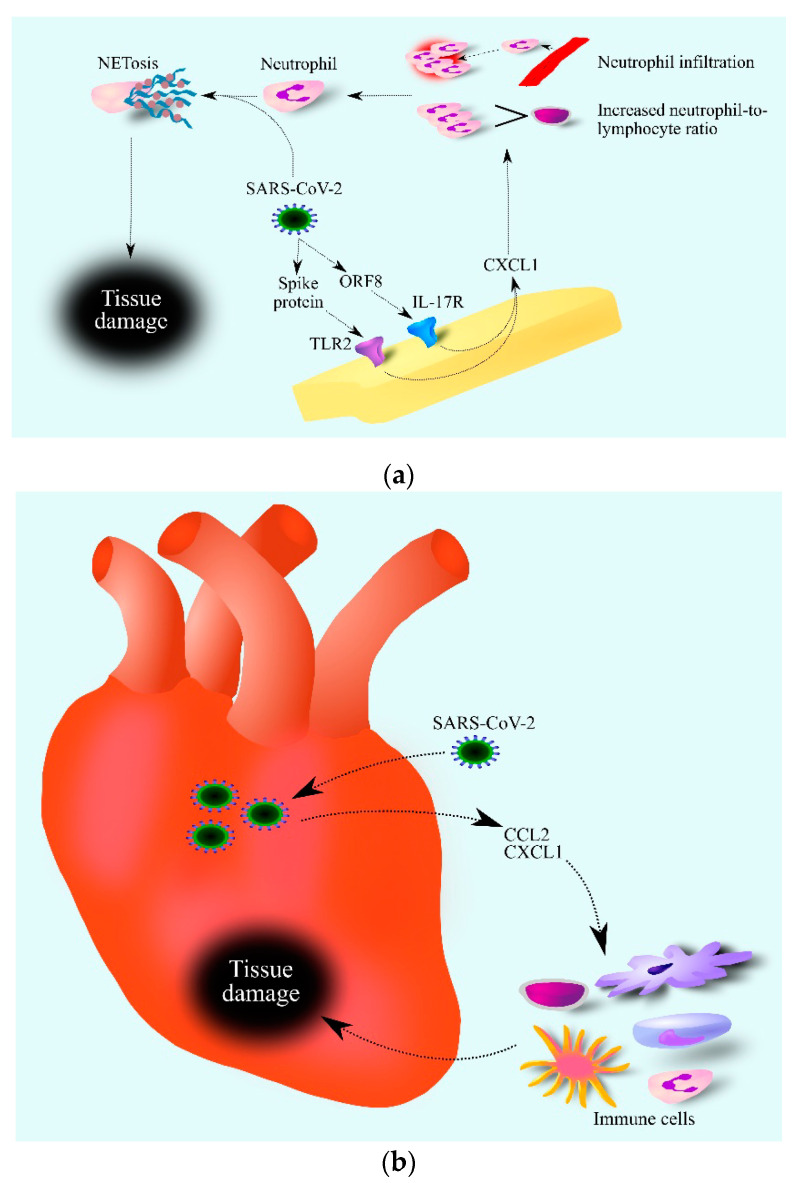
Role of CXCL1 in the course of COVID-19. SARS-CoV-2 increases CXCL1 expression (a). This is associated with TLR2 activation by the SARS-CoV-2 spike protein and IL-17R activation by the SARS-CoV-2 ORF8 protein. Increased levels of CXCL1 lead to an increase in the number of neutrophils in the blood and infiltration of infected tissue by these cells. SARS-CoV-2 then causes the activation of neutrophils which causes NETosis, i.e., regulated cell death dependent on the formation of NET. SARS-CoV-2 damages the heart (b) by increasing the expression of chemokines such as CCL2 and CXCL1. This causes the infiltration of the heart by various immune cells, which contribute to heart tissue damage. Abbreviations: CCL2–CC motif chemokine ligand 2; CXCL1–CXC motif chemokine ligand 1; NETosis–neutrophil extracellular traps formation; IL-17R– interleukin-17 receptor; ORF8–open reading frame 8; TLR2–toll-like receptor 2.

**Figure 3 ijms-24-00205-f003:**
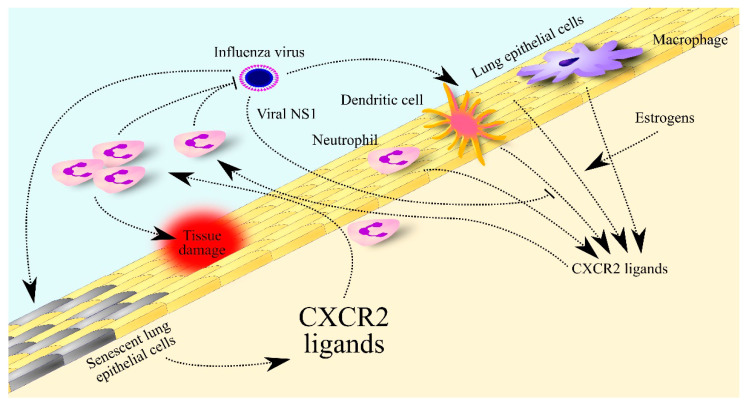
Significance of CXCR2 ligands in influenza. The influenza virus increases the expression of CXCR2 ligands in lung epithelial cells, dendritic cells, neutrophils and macrophages. This expression is decreased by viral non-structural protein 1 (NS1) protein but increased by estrogen in women. The elevated levels of CXCR2 ligands lead to the recruitment of neutrophils to the lungs to fight the virus. In older adults, lung epithelial cells secrete many more CXCR2 ligands in response to influenza virus infection, which leads to their much higher levels in the lungs and much greater infiltration of the lungs by neutrophils than in young people. The excessive number and activity of these cells contributes to lung tissue damage.

**Figure 4 ijms-24-00205-f004:**
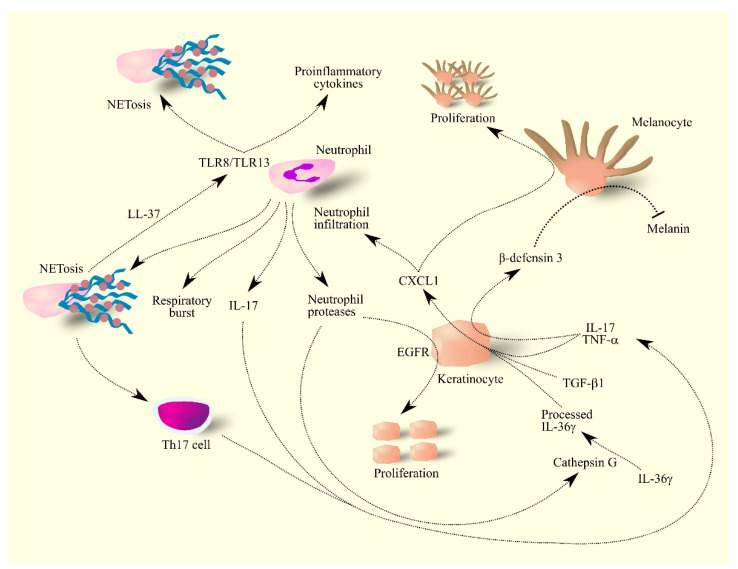
Importance of CXCL1 in molecular mechanisms in psoriasis. Keratinocytes under the influence of IL-17, TNF-α, TGF-β and IL-36γ produce CXCL1. These cells also produce β-defensin 3 under the influence of IL-17 and TNF-α. CXCL1 increases the proliferation of melanocytes in psoriasis lesions—these lesions have much less melanin than healthy skin because β-defensin 3 inhibits melanin production by melanocytes. CXCL1 also causes the infiltration of neutrophils into psoriasis lesions, where they produce IL-17 and secrete reactive compounds. Neutrophils also secrete proteases that activate ligands for EGFR; the activation of this receptor on keratinocytes causes the proliferation of these cells. The secretion of protease cathepsin G by neutrophils causes the processing of IL-36γ, an active form of IL-36γ. Neutrophils also produce NET which increase the activation of Th17 cells and also form complexes with LL-37 which activates other neutrophils and thus increases their production of proinflammatory cytokines and NET formation by other neutrophils. Abbreviations: CXCL1–CXC motif chemokine ligand 1; EGFR–epidermal growth factor receptor; IL-17–interleukin-17; IL-36γ–interleukin-36γ; NETosis–neutrophil extracellular traps formation; TGF-β–transforming growth factor β; Th17 cell–T helper type 17 cell; TLR8–toll-like receptor 8; TLR13–toll-like receptor 13; TNF-α–tumor necrosis factor α.

## Data Availability

Not applicable.

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
