# Peer review of "The Potential Importance of CXCL1 in the Physiological State and in Noncancer Diseases of the Cardiovascular System, Respiratory System and Skin"

_ijms, 2022, doi:10.3390/ijms24010205_

Round 1

Reviewer 1 Report

In this review article the authors presented the role of the CXCL1 in a number of major non-cancer diseases of the cardiovascular, respiratory systems, and the skin. CXCL1 is a chemokine with a CXCR2 its major receptor and is implicated in migration and infiltration of neutrophils to the sites of high expression. The authors introduce some examples of major diseases in the former three mentioned body systems where the CXCL1 play a role in their pathogenesis e.g atherosclerosis and chronic ischematic heart disease; bronchial asthma and COVID-19; and psoriasis and sunburn. The authors highlight the need for more researches which should include animal models to study the human CXCL1 particularly with the difference in the CXCR2 ligand systems between humans and rodents. Another direction of the researches are therapeutics approaches which target the chemokine and thus may inhibit the disease progression.   

Minor comments

It would more valuable if the authors include more figures like figures 1 and 2 illustrating the role of CXCL1 in other major non-cancer disease of the respiratory systems.

Author Response

Rev 1

  1. Minor comments

It would more valuable if the authors include more figures like figures 1 and 2 illustrating the role of CXCL1 in other major non-cancer disease of the respiratory systems.

According to Reviewer’s recommendations, 2 figures have been added.

Reviewer 2 Report

The authors have provided a review of the potential involvement of CXCL1 in a number of cardiovascular diseases. In general, the review is well written and broad in scope. The figures are useful. The main weakness is that, in spite of the title, they do not really address the importance of CXCL1 in these diseases. All of these, particularly sepsis, are very complex diseases that involve a complicated web of pro- and anti-inflammatory cytokines as well as chemokines. Many of these have been shown to be important by demonstrating that deletion or inhibition significantly affects the outcome of the disease (positively or negatively). Citations of such studies are absent in this review. 

The significance of the report is further decreased (as acknowledged by the authors) by the substantial differences between the murine, rat and human forms of the chemokine. As such, the report provides a sound collection of information, but the overall significance is modest at best.

Author Response

Rev 2

The authors have provided a review of the potential involvement of CXCL1 in a number of cardiovascular diseases. In general, the review is well written and broad in scope. The figures are useful. The main weakness is that, in spite of the title, they do not really address the importance of CXCL1 in these diseases. All of these, particularly sepsis, are very complex diseases that involve a complicated web of pro- and anti-inflammatory cytokines as well as chemokines. Many of these have been shown to be important by demonstrating that deletion or inhibition significantly affects the outcome of the disease (positively or negatively). Citations of such studies are absent in this review.

The significance of the report is further decreased (as acknowledged by the authors) by the substantial differences between the murine, rat and human forms of the chemokine. As such, the report provides a sound collection of information, but the overall significance is modest at best.

We agree with the reviewer that the paper lacks information about studies that have investigated the consequences of either knockout or inactivated human CXCL1 on  in vivo models of various diseases. This is due to the lack of a mouse model of human CXCR2 ligands. The CXCR2 ligand system differs between mouse and human. It is not possible to unequivocally equate a given CXCR2 ligand in mouse with its human counterpart. For this reason, we addressed this problem indirectly. We looked for information on whether patients with a particular disease have an upregulation of CXCL1 either in the blood or in the affected tissue. If this was the case, we looked for information on mouse/rat models of the disease in question. Usually, the results showed upregulation/expression of CXCR2 ligands in laboratory animals. The CXCR2 ligand in humans and mice has the same properties. It activates CXCR2. For this reason, we described the disease mechanisms discovered in mice as those most likely to occur in humans. Most often, the quoted mouse studies involved either a CXCR2 inhibitor or a gene knockout of the most important CXCR2 ligand, which is the reviewer's recommendation. Nevertheless, we have added a description of the problem: the differences in the CXCR2 ligand system between humans and laboratory animals.

Reviewer 3 Report

This is an original topic that will be of interest to the readers of the journal. It is generally well written and structured. The manuscript is clear and straight to the point. I have provided few minor revisions below.

Line 99-101: merit rephrasing

Line 111: missing reference, and please provide the full name for KC = keratinocyte-derived chemokine

Line 127: remove source: own elaboration

Line 144: add (Figure 1)

Line 159 -160: the authors are kindly requested to elaborate more on how macrophages are crucial for induction of hypertension and to provide evidence on how a similar mechanism may occur in hypertensive patients.

The same comment applies to the next lines (161- 166)

Line 146: the authors stated that Elevated blood levels of CXCL1 have been observed in patients with atrial fibrillation, but have elaborated on it.

Line 245: missing reference

Lines 250-254: merit further elaboration

Lines 264-268: merit further elaboration

Line 306: increases

Line 388: missing reference

Itchy skin merit further elaboration to elucidate the role of CXCL1

Some references must be updated.

Author Response

Rev 3

This is an original topic that will be of interest to the readers of the journal. It is generally well written and structured. The manuscript is clear and straight to the point. I have provided few minor revisions below.

Line 99-101: merit rephrasing

Line 111: missing reference, and please provide the full name for KC = keratinocyte-derived chemokine

Line 127: remove source: own elaboration

Line 144: add (Figure 1)

Line 159 -160: the authors are kindly requested to elaborate more on how macrophages are crucial for induction of hypertension and to provide evidence on how a similar mechanism may occur in hypertensive patients.

The same comment applies to the next lines (161- 166)

Line 146: the authors stated that Elevated blood levels of CXCL1 have been observed in patients with atrial fibrillation, but have elaborated on it.

Line 245: missing reference

Lines 250-254: merit further elaboration

Lines 264-268: merit further elaboration

Line 306: increases

Line 388: missing reference

Itchy skin merit further elaboration to elucidate the role of CXCL1

Some references must be updated.

The manuscript has been revised according to the reviewer's recommendations. The article has been updated using the latest articles.

Reviewer 4 Report

Korbecki et al. present a comprehensive review of CXCL1, and its receptor, CXCR2, in normal and disease states involving the skin, respiratory and cardiovascular systems. The modulation of the inflammatory response mediated by neutrophils in settings such as asthma, chronic obstructive pulmonary disease (COPD), chronic rhinosinusitis, coronavirus disease 2019 (COVID-19), influenza, lung transplantation and 23 ischemic-reperfusion injury, tuberculosis, atherosclerosis, chronic ischemic heart disease, hypertension, sepsis including sepsis-associated encephalopathy and sepsis-associated acute kidney injury, wound healing, psoriasis, sunburn and xeroderma pigmentosum in the 22 page work.

In general, this paper is an easy read, with the Introduction brief but specific and the search strategy standard and straight forward. Each section is focused but sufficiently detailed to leave the reader with a clear understanding of the concepts and paradigms presented.

With regard to the illustrations, the black lettering in the red ovals in figure 2 are difficult to read. Please improve the resolution of both figures to improve readability.

Lastly, please provide a Conclusion section that brings the messages together in a meaningful manner that includes a key concept or two from Section 6.

Author Response

Rev 4

Korbecki et al. present a comprehensive review of CXCL1, and its receptor, CXCR2, in normal and disease states involving the skin, respiratory and cardiovascular systems. The modulation of the inflammatory response mediated by neutrophils in settings such as asthma, chronic obstructive pulmonary disease (COPD), chronic rhinosinusitis, coronavirus disease 2019 (COVID-19), influenza, lung transplantation and 23 ischemic-reperfusion injury, tuberculosis, atherosclerosis, chronic ischemic heart disease, hypertension, sepsis including sepsis-associated encephalopathy and sepsis-associated acute kidney injury, wound healing, psoriasis, sunburn and xeroderma pigmentosum in the 22 page work.

In general, this paper is an easy read, with the Introduction brief but specific and the search strategy standard and straight forward. Each section is focused but sufficiently detailed to leave the reader with a clear understanding of the concepts and paradigms presented.

With regard to the illustrations, the black lettering in the red ovals in figure 2 are difficult to read. Please improve the resolution of both figures to improve readability.

Lastly, please provide a Conclusion section that brings the messages together in a meaningful manner that includes a key concept or two from Section 6.

As recommended by the reviewer, the figure has been changed. Also, conclusions have been added.

Round 2

Reviewer 2 Report

The authors have clearly acknowledged the difficulties in defining clear roles for CXCL1 in human disease because of differences with mouse chemokines and have provided additional details which greatly improve the paper. However, they have still not clearly defined importance.   I would recommend revising the title to "The potential importance of CXCL1". 

Author Response

Rev.2

The authors have clearly acknowledged the difficulties in defining clear roles for CXCL1 in human disease because of differences with mouse chemokines and have provided additional details which greatly improve the paper. However, they have still not clearly defined importance.   I would recommend revising the title to "The potential importance of CXCL1". 

The title has been changed to:  “The importance of CXCL1 in the physiological state and in noncancer diseases of the cardiovascular system, respiratory system and skin”
